# Burnout, Work Engagement and Other Psychological Variables During the COVID-19 Pandemic Among Nursing Students with Clinical Experience: A Pre–Post Study

**DOI:** 10.3390/healthcare13192446

**Published:** 2025-09-26

**Authors:** María José Membrive-Jiménez, Almudena Velando-Soriano, Luis Albendín-García, Guillermo A. Cañadas-De la Fuente, José L. Gómez-Urquiza, Gustavo R. Cañadas-De la Fuente

**Affiliations:** 1Faculty of Health Sciences of Ceuta, University of Granada, 51005 Ceuta, Spain; jlgurquiza@ugr.es; 2Instituto de Investigación Biosanitaria ibs.GRANADA, 18012 Granada, Spain; lualbgar1979@ugr.es; 3San Cecilio Clinical University Hospital, 18007 Granada, Spain; almudena.velando.sspa@juntadeandalucia.es; 4Casería de Montijo Health Center, Granada, Metropolitan District, Andalusian Health Service, 18013 Granada, Spain; 5Faculty of Health Sciences, University of Granada, 18012 Granada, Spain; gacf@ugr.es; 6Brain, Mind and Behaviour Research Center (CIMCYC), Campus Universitario de Cartuja, University of Granada, 18012 Granada, Spain; 7Department of Didactics of Mathematics, Faculty of Education, Campus Universitario de la Cartuja, University of Granada, 18012 Granada, Spain; grcanadas@ugr.es

**Keywords:** nursing students, academic burnout, anxiety, COVID-19, depression, clinical training

## Abstract

**Aims**: To analyze the psychological impact of the COVID-19 pandemic on nursing students who had just completed their first period of clinical placement and compare these results with a study previously conducted on the same students during their university education. **Design**: A pre–post design was used. **Methods**: Students who had already participated in a previous (February 2021) related study were sent a follow-up questionnaire (response rate = 52.8%) at the end of their clinical placement training period (June 2021). Descriptive analyses of the study variables were conducted, and burnout levels were estimated after the students had completed their clinical placement. Predictive models for the three dimensions of burnout were then obtained using multiple linear regression. **Results**: The study results suggest that a high proportion (47.2%) of nursing students who performed their first clinical placements during the COVID-19 pandemic experienced high levels of burnout. However, engagement was a protective factor against fear of COVID-19, anxiety, neuroticism, emotional exhaustion and reduced personal accomplishment. **Conclusions**: Nursing students who completed their first clinical placements during the COVID-19 pandemic were more likely to exhibit high levels of burnout and showed significant changes in their psychological dimensions. A risk profile should be established to identify the nursing students most vulnerable to developing high levels of burnout.

## 1. Introduction

Burnout syndrome, or occupational burnout, has long been studied. In one of the earliest papers in this respect, the German psychiatrist Herbert Freudenberger described it as a pathology characterized by a gradual loss of energy leading to exhaustion and demotivation in the workplace [1]. More recently, Ramírez-Elvira et al. [2] concluded that burnout has a direct impact on the work environment and on relationships among coworkers.

Burnout syndrome is associated with chronic, long-term work-related stress, normally characterized by three dimensions: emotional exhaustion (EE), depersonalization (D) and low personal accomplishment (PA). These three dimensions are examined and reflected in the Maslach Burnout Inventory (MBI), a standardized measurement instrument developed by Maslach and Jackson in 1986 [3].

In addition, models have been used to study the relationships between certain risk factors and burnout syndrome. There are different models that perform a developmental analysis, which is very useful in follow-up studies. One that has been used recently in studies conducted on nurses is the Golembiewski and Munzenrider model [4]. This model states that in phase 1, burnout levels are low in all three dimensions, in phases 2 to 7 the levels increase progressively, and in phase 8 the highest score is reached in all dimensions. This allows nurses to be classified according to the phase of overall burnout development they are in [5].

Initially, healthcare professionals were thought to be the workers most vulnerable to developing burnout syndrome. This risk is especially high among nurses and physicians [6] due to the close caregiving relationship they maintain with patients and their families. Depending on their patients’ health status, these professionals may become emotionally overinvolved [7].

Since its appearance, numerous measuring instruments based on the MBI have been validated and adapted to different professional categories, such as state security forces, truck drivers, nurses [8] and athletes [9].

Beyond the occupational context, burnout syndrome has also been identified in academic settings; for example, if students believe they cannot meet the challenges entailed in their education, they may adopt a negative or apathetic attitude, impairing the ability to complete their studies. Academic burnout is particularly problematic for university students, who often experience changing educational environments, geographic locations and teaching methods [10]. Furthermore, these students sometimes rely on a scholarship to complete their studies, the amount of which varies according to the grades obtained in each academic year. These issues, combined with academic pressure, the effort required to adapt to a new environment and the need to meet the financial cost of university studies, can trigger high levels of academic stress [11]. If it persists, this stress can provoke burnout syndrome, reduce academic performance and heighten the risk of dropout [12]. In response to these concerns, the Maslach Burnout Inventory-Student Survey (MBI-SS) was developed to measure academic burnout [13].

Nurses are among the professionals most susceptible to developing burnout and are vulnerable to the syndrome from the early stages of their university education [7]. This risk continues through their professional training, during which stressful situations commonly affect nurses’ emotional level and can lead to a deterioration in the care relationship with patients. In addition, theoretical education at university must be combined with clinical placement, an obligation that limits the time available for private study [14].

Because of this, as can be seen in other research, it is important to consider and promote work engagement. This concept is essential to ensure that nurses can provide quality care, as they are the backbone of any healthcare system [8,15]. Also important are questions such as the students’ satisfaction with their teachers and with the instruction provided, and their own commitment to the nursing program. These factors can directly influence the development or otherwise of academic burnout [16].

Studies conducted on Spanish nurses confirm that burnout rates reached up to 70% during the pandemic [17]. According to a national study conducted on Spanish university students, approximately 50% needed psychological support because of COVID-19 [18]. However, starting in 2020, during their clinical rotations, nursing students were already exposed to a heavy workload, as well as uncertainty about the future spread of COVID-19 [19]. These third- and fourth-year nursing students were exposed to psychological distress and secondary trauma from the global pandemic during their college education, which may have included isolation, lack of in-person communication, and academic compromise [20,21]. In addition, students had to adapt to new clinical settings, acquire unfamiliar clinical skills, and manage the additional workload under adverse conditions. This could lead to compassion fatigue and secondary post-traumatic stress [22].

For this reason, the research question guiding this study was whether undertaking their first clinical placement during the COVID-19 pandemic had a greater psychological impact on nursing students. Then, we compared the study results with those obtained in earlier research conducted with the same students during their theoretical training at the university [8]. Therefore, due to stress, perceived lack of support, academic deficiencies, and consequent mental health problems, it is plausible that third- and fourth-year nursing students would develop academic burnout [23].

In view of these considerations, the aim of this study was to analyze the influence of the COVID-19 pandemic on nursing students, as an additional stressor experienced during their first practical training [24]. Therefore, the evolution of burnout levels, engagement, and other psychological variables was studied following the clinical experience acquired by students during their first period of healthcare internships.

## 2. Materials and Methods

### 2.1. Study Design and Variables

A pre–post design was used [25]. At the beginning of the second semester of 2021, during the COVID-19 pandemic, the researchers contacted the students who were scheduled to complete their first clinical placement for the Nursing Degree course and invited them to participate in the study. To avoid potential bias, the same members of the research team provided initial information and later collected the data—always at the beginning of the mandatory reinforcement seminars that were held.

The following independent variables were considered: health status, fear of COVID-19, personality dimensions (neuroticism, extraversion, agreeableness and conscientiousness), anxiety, depression, and the three dimensions of engagement (dedication, absorption and vigor). The dependent variable measured was burnout. All these variables are conceptually defined as established by the instruments used to measure them, which are included in the “Instruments” section.

### 2.2. Participants

The criteria applied for inclusion in this research were that participants should be nursing students at the University of Granada, have recently completed their first clinical placement, and also have participated in the first part of the research, immediately prior to this internship. In the first part of the study, all nursing students who were about to begin their clinical placement were invited to participate (212 third-year nursing students from the University of Granada). Non-university nursing students and those studying for other university degrees were excluded. The final sample of participants consisted of 112 eligible students, for whom data obtained before and after their first clinical placement were available (response rate = 52.8%). The mean age of participants was 22.21 years (SD = 1.73), 86.6% were female, 54.5% had completed their internships in primary care departments and 45.5% had done so in hospital care.

### 2.3. Procedures

The data obtained were collected on a web platform at two points: the first, before the clinical practice period, in February 2021 [8], and the second in June 2021, when the students had completed their clinical practice period. The information was collected in person during seminars that complemented the healthcare placements. This was the only time that students attended the university in person. A web platform was used for this purpose, which each student filled out individually. All participants signed an informed consent form and were guaranteed the confidentiality and anonymity of the data provided.

### 2.4. Instruments

The participants each completed a questionnaire (142 items), providing sociodemographic data and information about their own health status and that of their immediate family members. Among these questions, the student nurses were asked whether they or someone close to them had contracted COVID-19, and about the physical and psychological consequences to them of the pandemic. The following validated measurement instruments were applied:

The Fear of COVID-19 Scale (FCV-19S), as adapted for use with a population of Spanish students [26]. This instrument was created by Ahorsu. et al. [27]; consists of seven items, each scored on a five-point scale and has good psychometric properties. A higher total score reflects greater fear of COVID-19.

Personality traits were measured according to four of the five dimensions of the Spanish version of the NEO Five-Factor Inventory (NEO-FFI), namely neuroticism, extraversion, agreeableness and conscientiousness [28]. The Openness dimension is not included because it is not a significant predictor of burnout and to minimize the number of items in the questionnaires. This instrument consists of 48 items scored using a five-point Likert response format. Each sub-scale (dimension) contains twelve items. Higher scores in the subscales indicate a higher degree of presenting these personality characteristics. The Cronbach’s alpha coefficients in the sample of participants were 0.838 for neuroticism, 0.844 for extraversion, 0.734 for agreeableness, and 0.826 for conscientiousness.

The Hospital Anxiety and Depression Scale (HADS) [29] as adapted for use with a Spanish population, with two subscales: anxiety (seven items) and depression (seven items) [30]. These fourteen items are scored on a four-point Likert scale (from 0 to 3 points). These scales present score ranges that indicate the probable absence (0 to 7), possible presence (8 to 10), and probable presence (11 to 21) of clinically significant degrees of mood disorders. Higher scores reflect a higher level of depression and anxiety. The internal consistency of the two subscales is 0.802 for anxiety and 0.784 for depression.

Engagement was measured with the Utrecht Work Engagement Scale (UWES) [13]. This instrument examines 24 items, scored on a seven-point response scale, considering the following dimensions of work engagement: Absorption (AB): full concentration and placid immersion in one’s own tasks; Dedication (DE): commitment to one’s own tasks, together with feelings of importance and enthusiasm; and Vigor (VI): energy and mental resilience. In all subscales, higher scores imply a higher level of engagement. The values obtained were 0.835 (in the vigor dimension), 0.868 (in the dedication dimension), and 0.640 (in the absorption dimension).

Academic burnout was evaluated using the Granada Burnout Questionnaire for University Students (“Questionnaire de Burnout Granada” CBG-US) [31]. This instrument is composed of 26 items scored on a five-point Likert scale. The CBG-US measures three dimensions of burnout syndrome: EE, D and PA. Higher scores in EE and D, and lower scores in PA, suggest higher levels of burnout. The Cronbach’s alpha coefficients in the sample of participants were 0.873 in EE, 0.878 in D, and 0.894 in PA. Information regarding the items, scoring, validation, and cut-off points of the CBG-US can be requested from the authors of this study.

### 2.5. Data Analysis

Statistical analyses were performed using IBM SPSS Statistics, version 25 (IBM Corp., Armonk, NY, USA) software. In this process, descriptive analyses were conducted of the study variables, and burnout levels were estimated after the students had completed their clinical practices. A correlation analysis was performed to verify that the relationships between variables were as they should be in order to confirm the validity of the instruments. Pre–post differences were then estimated, taking into account the pandemic-related conditions reported by the study participants and the type of internships completed. These comparisons were made using Student’s *t*-test. For this purpose, the basic assumptions of the technique were also tested. One such was the assumption of homogeneity of variances. If this condition was not met, the Welch approximation was used. Predictive models for the three dimensions of burnout were then obtained using multiple linear regression, using the remaining variables as predictors. The predictor variables in the regression models were selected based on their theoretical relevance according to the scientific literature. The assumptions of normality, linearity, collinearity and heteroscedasticity were each examined. To avoid potential confounding factors, the variables age, number of children, and gender were considered. The first two were found to generate collinearity and were therefore eliminated from the analysis. Similarly, the gender variable was eliminated. In this case, complete stratification cannot be performed in the regression analysis due to the low percentage of men in the sample.

Finally, the differences between the results obtained before and after clinical practice, for the burnout dimensions, the degree of engagement and the psychological variables considered, were determined and analyzed.

### 2.6. Ethical Considerations

The study was approved by the Ethics Committee of the University of Granada (2416/CEIH/2021) on 17 November 2021, and the ethical considerations of the Declaration of Helsinki [31] were complied with at all times. The data were processed in accordance with the provisions of Act 3/(5 December 2018), on Personal Data Protection and guarantee of digital rights.

## 3. Results

### 3.1. Burnout Levels and Related Variables

Among the study participants, 53.6% reported having had COVID-19, although 7.1% did not respond to this question. In 78.6% of cases, a family member had experienced the disease, while in 8.9% of cases, they had not (12.5% of participants did not respond to this question). Further, 16.1% of students reported the death of someone close to them due to COVID-19. Long-term consequences of the pandemic, physical and psychological, were experienced by 37.5% and 67.9% of respondents, respectively.

Information on anxiety and depression was obtained following the recommendations of the HADS authors [29]. In detail, 31.5% and 7.2% of the students were classified as borderline anxiety and depression, respectively, while 18% and 10.8%, respectively, were considered probable cases of anxiety and depression.

Table 1 presents the descriptive statistics and correlations obtained for the study variables. The correlation matrix was obtained to confirm evidence of concurrent validity of the burnout measurement questionnaire. Overall, the correlation results reflect that both the intensity and direction of the relationships between the variables coincide with those reported in previous research. This indicates that the patterns of association between burnout, engagement, and the psychological variables studied remain consistent with the evidence found in scientific literature.

The correlation analysis between the study variables revealed moderate, statistically significant correlations in almost all cases (see Table 1), in the expected sense.

The participants were classified according to the severity of their burnout, combining the results obtained for each dimension of the syndrome according to the model proposed by Golembiewski et al. [32]. This approach allows us to distinguish the degrees to which students are affected by burnout syndrome, through the combination of its three dimensions (EE, D, PA). Assignment to each phase is made by considering whether the scores for each dimension are low or high. If a subject is in phase 1, 2, or 3, they have low burnout; if they are in phase 4 or 5, they have medium burnout; and if they are in phase 6, 7, or 8, they have high burnout (see Table 2).

This analysis showed that 47.2% of the participants were classified as stages 6, 7 or 8 (corresponding to high levels) of burnout, with the remainder presenting moderate or low levels of the disorder. This proportion is higher than that recorded before the students’ internships [8].

### 3.2. Levels of Burnout and Engagement According to Gender, Type of Internship and COVID-Related Variables

Hypothesis tests were conducted to identify possible differences between groups of participants, for each of the variables considered. These groups were composed, on the one hand, according to the type of internships performed (primary care or hospital care), and on the other, by gender. In the first respect, there were no statistically significant differences between the students who completed their clinical placements in primary care versus those who completed them in hospitals, for any of the variables considered.

However, significant differences were found between male and female participants with respect to the following variables: AG (t(109) = −3.203, *p* = 0.002), CO (t(110) = −3.347, *p* = 0.001) and DE (t(17.489) = −2.668, *p* = 0.016), with women scoring higher than men in every case. By contrast, men scored significantly higher than women for D (t(110) = 3.067, *p* = 0.003) and DP (t(109) = 4.264, *p* < 0.001).

Hypothesis tests were also performed to detect differences between the variables according to the impact produced by COVID-related factors. These tests revealed no significant differences between the students who had acquired COVID-19 and those who had not, for any of the variables considered. On the other hand, the students who reported that a relative or close person had been infected with COVID-19 scored significantly higher for PA (t(87) = −2.216, *p* = 0.029), AG (t(85) = −2.777, *p* = 0.007) and EX (t(86) = −2.788, *p* = 0.007). Furthermore, these students reported significantly lower values for NE (t(87) = 3.471, *p* = 0.001), AN (t(87) = 2.149, *p* = 0.034), DP (t(87) = 2.097, *p* = 0.039) and FCV (t(75) = 2.492, *p* = 0.015). Finally, significant differences were found in terms of AG (t(97) = −2.585, *p* = 0.011) and VI (t(99) = −2.133, *p* = 0.035), scoring higher the students who stated that someone close to them who had died from COVID-19, and those for whom this was not the case.

### 3.3. Risk Factors and Predictive Models of Burnout

Three predictive models were obtained, one for each dimension of burnout, from the remaining variables considered. The values obtained were significant in every case: *F*(3, 85) = 12.992, *p* < 0.001, *R*^2^*_Adj_* = 0.314 for EE; *F*(2, 85) = 36.163, *p* < 0.001, *R*^2^*_Adj_* = 0.46 for D; and *F*(4, 85) = 28.657, *p* < 0.001, *R*^2^*_Adj_* = 0.574 for PA (see Table 3).

The regression analyses performed reveal that the variables VI, AN, and EX were identified as significant predictors of the EE dimension. Likewise, the variables DP and AG were found to be predictors of the D dimension. Finally, the variables CO, DE, DP, and AB were relevant predictors of PA among students. It is important to note that FCV was not found to be a significant predictor in any of the burnout dimensions evaluated. This implies that personal and personality factors have a greater influence than the direct emotional impact of the pandemic on the development of academic burnout.

### 3.4. Pre–Post Comparison of the Study Variables

The pre–post analysis (with respect to the beginning and end of the clinical practice/internship period) revealed significant differences for EE (t(86) = 4.586, *p* < 0.001) and PA (t(88) = −4.826, *p* < 0.001), with less EE and higher levels of PA on finishing the internship period. Moreover, levels of N (t(87) = 3.464, *p* = 0.001), AN (t(87) = 2.505, *p* = 0.014) and FCV(t(76) = 6.430, *p* < 0.001) were significantly lower at the end of this period.

## 4. Discussion

In this study, we analyzed the impact produced by the COVID-19 pandemic on the mental health of nursing students who had completed their first clinical placement and compare the results with corresponding measurements obtained before the internship.

The study results, indicating a significant relation between job stress and personality characteristics, corroborate those of previous research in this line [8]. Specifically, our findings suggest that individual factors of emotional well-being, such as anxiety and depression, together with dimensions of engagement, are crucial aspects of burnout syndrome and should be considered in this context [33]. Among the study participants, 18% were classed as probable cases of anxiety and 10.8% as liable to depression. With respect to the earlier study performed, these results differ for anxiety (30.8%) but are similar for depression (9.2%) [8]. From this information, we deduce that students develop coping strategies during their clinical practice, making them better equipped to respond to stressful situations than during their pre-internship period [16]. These results are in line with those obtained in a previous meta-analysis of nursing students during the pandemic [34].

One of the major findings of this study is the greater numbers of students who presented a high level of burnout at the end of clinical practice (47.2%), compared to their pre-internship period (37.8%) [8]. These data are consistent with data provided in other studies, which analyzed the nursing student population in countries on all continents (China, USA, Brazil, Portugal, South Africa, etc.). Prevalence ranged from 19% to 41% depending on the country and its education system [10]. In the case of registered nurses, the global prevalence prior to the pandemic had been between 4.7% and 13.7%, or even up to 30%. However, after the pandemic, the figures rose to an average of 59.5%, especially in Europe and Africa. Countries such as Spain and Italy have reached prevalences of 70% and 77%, respectively [17,35,36].

This increase is consistent with the literature, according to which nursing students who began their clinical placements during the COVID-19 pandemic were particularly vulnerable to developing burnout. The transition from an academic to a clinical setting, together with the new learning methodology required in dealing with real patients in the context of the COVID-19 pandemic, could have exacerbated stress factors such as emotional burden and exposure to traumatic situations [37,38]. Our study data are also consistent with those obtained in research conducted before the COVID-19 pandemic, which reports a direct association between the development of burnout and the performance of clinical practice [16], especially in students who have not developed resilience skills, those who are not strongly committed to their university studies [39], those who are not satisfied with the internship experience and those who encounter problems during this period [16].

Regarding the different dimensions of burnout, the participants in our study reported less EE and higher levels of PA compared to the analysis carried out before the internship [8], together with significantly lower levels of neuroticism and fear of COVID [8]. These results are consistent with the Burnout-Engagement Model proposed by Leiter and Maslach [40], according to which the care experience with real patients and the acquisition of practical skills can provide students with a greater sense of motivation and satisfaction, counteracting the negative effects of burnout.

The female participants scored significantly higher in the dimensions of agreeableness, conscientiousness and dedication, while their male counterparts presented higher levels of depersonalization and depression. This finding could be associated with the fact that women who perform healthcare tasks tend to have higher levels of engagement (in the form of enthusiasm and dedication) [41]. By contrast, men in this situation may experience a psychological barrier to expressing their emotions in the workplace, which might provoke both depression and some depersonalization in their therapeutic relationships with patients.

Analysis of the impact of COVID-19 showed that the students whose families were directly affected presented higher levels of personal accomplishment, agreeableness and extraversion, and lower ones of neuroticism, anxiety, depression and fear of COVID-19. These results suggest that the direct relationship with the illness of a close person could decrease the fear of COVID-19 [42], which in turn would have a positive impact on psychological well-being, promoting resilience skills and making COVID-19 a less potent stressor [23,43].

This study is subject to certain limitations that should be considered when interpreting the results presented. Firstly, the study design was pre–post, with limited follow-up of the nursing students taking part. Moreover, the sampling was not random, as the nature of the study meant that the same students participated in both instances (in the present case and in the previous one, reported by Cañadas de la Fuente et al. [8]), meaning that the questionnaire questions were already familiar to them. For these reasons, the results of this study should be taken with caution. Another limitation was the absence of a control group of students who had not experienced the COVID-19 pandemic, which would have allowed for a comparison between both groups, and the identification of causal relationships between psychological changes and the pandemic. Finally, it should be noted that the students were from a single university and may have exhibited a possible social desirability bias.

In future research, it would be useful to conduct a study of the coping strategies these students used to maintain their mental health and prevent the development of burnout during their clinical practice in a period characterized by the COVID-19 pandemic. In addition, strategies should be implemented to help students determine the presence and impact of negative psychological symptoms and thus prevent or alleviate the development of burnout. Several authors claim that interventions based on muscle relaxation exercises or behavioral teaching sessions focused on personal and professional development and improving coping skills would help prevent academic burnout [44]. Furthermore, fostering empathy among nursing students is essential to reducing levels of academic burnout. Encouraging empathy improves communication, teamwork, job satisfaction, and emotional resilience. All of this can be implemented through simulation-based learning policies, reflective exercises, and communication training [45].

## 5. Conclusions

The results of this study suggest that, by comparison with those obtained during the period of theoretical instruction, the nursing students who completed their first internships during the COVID-19 pandemic were more likely to exhibit high levels of burnout and evidenced significant changes in their psychological dimensions. However, in terms of engagement, the clinical practice period was featured by reduced emotional exhaustion, fear of COVID-19, anxiety and neuroticism, and a heightened sense of personal accomplishment.

In view of these findings, we believe it necessary to create a risk profile of nursing students to detect those most vulnerable to developing high levels of burnout. The study results we present show that personality traits and various psychological factors (such as anxiety, depression and engagement) are significant predictors of the dimensions of burnout syndrome in times of health crises such as the COVID-19 pandemic.

## Figures and Tables

**Table 1 healthcare-13-02446-t001:** Descriptive statistics and correlations for the study variables (n = 112).

V	1	2	3	4	5	6	7	8	9	10	11	12	13
M	25.92	11.7	38.95	33.9	43.76	42.77	45.34	13.95	8.34	13.64	10.33	7.42	4.67
SD	6.404	4.18	6.569	8.599	5.357	6.971	6.158	4.702	4.104	4.136	3.594	3.697	3.711
EE	1												
D	0.203 *	1											
PA	−0.611 **	−0.62 **	1										
NE	0.448 **	0.226 *	−0.454 **	1									
AG	−0.247 **	−0.512 **	0.589 **	−0.313 **	1								
EX	−0.117	−0.437 **	0.421 **	−0.418 **	0.344 **	1							
CO	−0.185	−0.492 **	0.532 **	−0.402 **	0.446 **	0.389 **	1						
FCV	0.138	−0.016	−0.047	0.309 **	0.099	−0.181	−0.062	1					
VI	−0.507 **	−0.368 **	0.528 **	−0.363 **	0.473 **	0.307 **	0.345 **	0.048	1				
DE	−0.35 **	−0.596 **	0.705 **	−0.235 *	0.59 **	0.348 **	0.401 **	0.03	0.538 **	1			
AB	−0.431 **	−0.423 **	0.603 **	−0.23 *	0.427 **	0.238 *	0.324 **	−0.057	0.729 **	0.644 **	1		
AN	0.451 **	0.263 **	−0.459 **	0.68 **	−0.325	−0.42 **	−0.43 **	0.461 **	−0.305 **	−0.324 **	−0.228 *	1	
DP	0.428 **	0.576 **	−0.634 **	0.48 **	−0.441 **	−0.424 **	−0.476 **	0.107	−0.389 **	−0.61 **	−0.42 **	0.622 **	1

V = Variable; M= Mean; SD = Standard deviation; EE = Emotional exhaustion (CBG); D = Depersonalization (CBG); PA = Personal accomplishment (CBG); NE = Neuroticism; AG = Agreeableness; EX = Extraversion; CO = Conscientiousness; FCV = Fear of COVID; VI = Vigor; DE = Dedication; AB = Absorption; AN = Anxiety; DP = Depression; 1 = Emotional exhaustion (CBG); 2 = Depersonalization (CBG); 3 = Personal accomplishment (CBG); 4 = Neuroticism; 5 = Agreeableness; 6 = Extraversion; 7, Conscientiousness; 8 = Fear of COVID; 9 = Vigor; 10 = Dedication; 11 = Absorption; 12 = Anxiety; 13 = Depression. * *p* < 0.05. ** *p* < 0.01.

**Table 2 healthcare-13-02446-t002:** Classification of the participants according to the phase model of Golembiewski et al. [32].

Phase	1	2	3	4	5	6	7	8
D	L	H	L	H	L	H	L	H
PA	H	H	L	L	H	H	L	L
EE	L	L	L	L	H	H	H	H
CBG %(n)	3.6 (4)	2.7 (3)	25.5 (28)	12.7 (14)	8.2 (9)	24.5 (27)	19.1 (21)	3.6 (4)

EE = Emotional exhaustion; D = Depersonalization; PA = Personal Accomplishment; CBG = Granada Burnout Questionnaire; L = Low; H = High.

**Table 3 healthcare-13-02446-t003:** Multiple regression models for each dimension of burnout syndrome.

Variable	B	SE	95% CI for *B*		*p*	β	t
			**LL**	**UL**			
EE							
Intercept	170.757	40.175	90.456	260.058	<00.001		40.253
VI	−0.578	0.132	−0.841	−0.315	<00.001	−0.411	−40.37
AN	0.631	0.163	0.307	0.955	<00.001	0.387	30.871
EX	0.198	0.082	0.035	0.361	0.018	0.244	20.414
D							
Intercept	190.737	20.953	130.866	250.608	<00.001		60.684
DP	0.551	0.1	0.353	0.749	<00.001	0.476	50.536
AG	−0.243	0.062	−0.367	−0.119	<00.001	−0.336	−30.9
PA							
Intercept	190.549	40.087	110.423	270.674	<00.001		40.784
CO	0.293	0.08	0.135	0.451	<00.001	0.32	30.686
DE	0.353	0.155	0.045	0.661	0.025	0.232	20.278
DP	−0.372	0.155	−0.68	−0.063	0.019	−0.219	−20.395
AB	0.317	0.156	0.007	0.627	0.045	0.188	20.031

*Note.* CI = Confidence Interval; LL = Lower Limit; UL = Upper Limit; SE = Standard Error; B = B Regression Coefficient; EE = Emotional exhaustion; D = Depersonalization; PA = Personal accomplishment; VI = Vigor; AN = Anxiety; EX = Extraversion; DP = Depression; AG = Agreeableness; CO = Conscientiousness; DE = Dedication; AB = Absorption.

## Data Availability

Data available under request to the corresponding author. The data supporting the findings of this study consist of individual-level responses containing sensitive and confidential information. For ethical and privacy reasons, the dataset is not publicly available.

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
