# Peer review of "Burnout, Work Engagement and Other Psychological Variables During the COVID-19 Pandemic Among Nursing Students with Clinical Experience: A Pre–Post Study"

_healthcare, 2025, doi:10.3390/healthcare13192446_

Round 1

Reviewer 1 Report

Comments and Suggestions for Authors

The article applies a pre-post design to assess burnout and engagement levels in nursing students before and after their first clinical internship during the COVID-19 pandemic. A sample of 112 students was evaluated using validated instruments, including the CBG-US, UWES, HADS, NEO-FFI, and FCV-19S. Results indicate that 47.2% of students experienced high burnout post-internship, with engagement acting as a mitigating factor. Regression models confirmed neuroticism, anxiety, and depression as significant predictors of burnout dimensions, while dedication and conscientiousness positively influenced personal accomplishment. Pre-post comparisons revealed reduced emotional exhaustion, anxiety, and neuroticism, and increased personal accomplishment after clinical practice. However, the study has methodological limitations, including a non-random sample and familiarization with the instruments. Improvements should include controlling for confounders in regression analysis, clarifying gender effects with interaction terms, and ensuring instrument blinding in longitudinal assessments. Overall, the findings support the relevance of psychological and personality factors in moderating burnout but require cautious interpretation due to design constraints.

Author Response

The article applies a pre-post design to assess burnout and engagement levels in nursing students before and after their first clinical internship during the COVID-19 pandemic. A sample of 112 students was evaluated using validated instruments, including the CBG-US, UWES, HADS, NEO-FFI, and FCV-19S. Results indicate that 47.2% of students experienced high burnout post-internship, with engagement acting as a mitigating factor. Regression models confirmed neuroticism, anxiety, and depression as significant predictors of burnout dimensions, while dedication and conscientiousness positively influenced personal accomplishment. Pre-post comparisons revealed reduced emotional exhaustion, anxiety, and neuroticism, and increased personal accomplishment after clinical practice.

Dear reviewer, thank you for your comments. Below we have described the changes made.

However, the study has methodological limitations, including a non-random sample and familiarization with the instruments.

We have already mentioned that at the end of the discussion.

 Improvements should include controlling for confounders in regression analysis, clarifying gender effects with interaction terms, and ensuring instrument blinding in longitudinal assessments.

The sample of participants consisted of all nursing students who completed their internship period, which was our population of interest. Although it is true that the total number of participants who were measured on two occasions was 112 students, participants who did not complete the questionnaires in both measurements were not included in the analysis.

To avoid undesirable effects from potential confounding factors (age, children, and gender), two steps were taken. In the regression analysis, it was verified that there was collinearity between the variables of age and number of children with the predictors included in the model; therefore, these variables were eliminated (a comment has been included in the study). With regard to gender, it was found that the regression models for the total sample are practically identical to the models obtained for the group of women. They were not included in the study because it would mean presenting two almost identical tables. Complete stratification cannot be performed with regression analysis in the group of men due to the low percentage of male participants. However, these results are those that truly correspond to our total sample and our target population; a very high percentage of nursing students are women.

In addition to the above, we have used inclusive language throughout the process to avoid inequalities. The aim was to promote gender mainstreaming.

With regard to blinding in the evaluation of the instruments, we have done several things. The team members themselves directed the application of the tests in each of the two measures collected. Additional analyses were performed to ensure the reliability and validity of the instruments. The correlation matrix between the variables was calculated to confirm the validity of the instruments used. The reliability of all instruments in the sample of participants was calculated.

All of the above aspects have been included in the corresponding sections of the study (sections 2.2, 2.3, 2.4, and in the discussion, within the limitations of the research).

Overall, the findings support the relevance of psychological and personality factors in moderating burnout but require cautious interpretation due to design constraints.

We have extended the limitations according to your instructions.

Reviewer 2 Report

Comments and Suggestions for Authors

The study entitled "Burnout and Professional Engagement during the COVID-19 Pandemic among Nursing Students with Clinical Experience: A Pre-Post Study" addresses an undeniably relevant and timely topic. Investigating burnout and engagement in nursing students during the COVID-19 crisis offers valuable insights, particularly given the extraordinary pressures placed on healthcare trainees during the pandemic. The research questions are well aligned with the broader discourse on mental health in healthcare education, and the longitudinal aspect of the design provides a welcome contribution to the existing body of literature.

However, it is somewhat surprising that the data on fear of COVID-19, personality, anxiety, and burnout were collected in 2021, yet the findings are only now being disseminated. While this delay is understandable given the disruptions caused by the pandemic itself, it does slightly reduce the immediacy of the study’s practical implications. In today’s largely post-COVID environment—where public attention has shifted away from pandemic-related concerns—the relevance of some findings may not be as pronounced as they would have been had they been published closer to the time of data collection. This is unfortunate, as the study could have had greater impact in guiding support strategies for students during the height of the crisis. Nevertheless, I welcome the publication of these findings and believe the article has potential to contribute meaningfully to the literature on student wellbeing and professional development in nursing.

At the same time, I believe the manuscript requires substantial revision to reach its full potential. Burnout is currently described in a relatively narrow manner, focusing on the classic tripartite dimensions of emotional exhaustion, depersonalisation, and reduced personal accomplishment. While this model remains foundational, I encourage the authors to integrate broader and more contemporary conceptualisations of burnout. Including alternative frameworks or recent developments in burnout research would enhance the theoretical depth of the study and increase its relevance to diverse audiences.

In addition, I noted the absence of a more explicit connection to the concepts of secondary traumatization, trauma transmission, and compassion fatigue—topics highly pertinent to nursing students with clinical exposure during a pandemic. Given the emotional toll of such experiences, these constructs deserve at least some attention in the discussion and should ideally be situated in the theoretical background. Their inclusion would offer a more comprehensive understanding of the psychological landscape encountered by student nurses in times of crisis.

Finally, I would like to point out that new theoretical perspectives are occasionally introduced later in the manuscript (e.g., in the results or discussion sections) rather than being clearly established in the theoretical framework. To improve clarity and coherence, it would be advisable to relocate these conceptual discussions to the background section, thereby ensuring a more logically structured narrative throughout the article.

In conclusion, while the study provides a relevant and important contribution to the field, its full value will only be realised through careful revision. I commend the authors for their efforts and encourage them to further develop the manuscript with these suggestions in mind.

More detailed feedback and comments:

  • I would like additional information on how the Spanish version of instruments has been provided. For example, did the authors provide the Spanish version of the Fear of COVID-19 scale themselves? How? Has this scale been validated and used in previous research yet? Is it consistent? (Cronbach Alpha score)
  • It is not clear why the research decided to use and measure only 4 of the 5 NEO-FFI dimensions?
  • For the differents instruments, I would also have liked some examples of items in the text.
  • Generally, I would appreciate some information about the validity and the reliability of the scales/instruments used in this study?

For the elaboration of regression models, I assume that the authors made the choice of a stepwise approach in order to obtain the best model. How did they make the choice for different variables. Are the different predictor variables introduced in the model according to the theoretical hypotheses of the researchers?

Explain Table 1 in words. A table should not be seen as a replacement of text or explanation. It helps to synthetize but is it is not the job of the reader to extricate the necessary information from the table(s). The numerous abbreviations make this task even more complicated.

Please consider a different way to present the results more clearly.

Table 2 mentions the model of Golembiewski? This is new for the reader since this model has not been presented previously in the theoretical section. As such, the is no information in a statement about stage 6 or 7 of the model. Also in the discussion section, there is nothing additional on this model.

Table 3 also needs more explanation in order to make the results understandable for the reader.

Discussion

What is the clinical relevance or usefulness of sentences like '... 18% were classed as probable causes of anxiety and 10,8% as liable to depression'?

The authors should make the difference between 'symptoms that may be indicative of ... (diagnosis in DSM-5 or ICD-11), symptomatology that may be an indication of... but these statements contain no relevant clinical information. Therefore, already in the methods section, it is important to mention WHAT EXACTLY is measured. It should also be presented that clearly in the results section. Therefore, both sections still need a lot of improvement.

Author Response

The study entitled "Burnout and Professional Engagement during the COVID-19 Pandemic among Nursing Students with Clinical Experience: A Pre-Post Study" addresses an undeniably relevant and timely topic. Investigating burnout and engagement in nursing students during the COVID-19 crisis offers valuable insights, particularly given the extraordinary pressures placed on healthcare trainees during the pandemic. The research questions are well aligned with the broader discourse on mental health in healthcare education, and the longitudinal aspect of the design provides a welcome contribution to the existing body of literature.

Dear reviewer, thank you for your comments.

However, it is somewhat surprising that the data on fear of COVID-19, personality, anxiety, and burnout were collected in 2021, yet the findings are only now being disseminated. While this delay is understandable given the disruptions caused by the pandemic itself, it does slightly reduce the immediacy of the study’s practical implications. In today’s largely post-COVID environment—where public attention has shifted away from pandemic-related concerns—the relevance of some findings may not be as pronounced as they would have been had they been published closer to the time of data collection. This is unfortunate, as the study could have had greater impact in guiding support strategies for students during the height of the crisis. Nevertheless, I welcome the publication of these findings and believe the article has potential to contribute meaningfully to the literature on student wellbeing and professional development in nursing.

The pandemic certainly caused many complications for the research projects that were underway at the time. In our case, there were many difficulties in collecting data for a broader longitudinal study. The project was completed in mid-2023, and analysis of the collected information began shortly thereafter.

At the same time, I believe the manuscript requires substantial revision to reach its full potential. Burnout is currently described in a relatively narrow manner, focusing on the classic tripartite dimensions of emotional exhaustion, depersonalisation, and reduced personal accomplishment. While this model remains foundational, I encourage the authors to integrate broader and more contemporary conceptualisations of burnout. Including alternative frameworks or recent developments in burnout research would enhance the theoretical depth of the study and increase its relevance to diverse audiences.

Thank you for your comment. We have expanded the information in the “introduction” with the model used. In addition, references to recent studies with the model described and used in this manuscript have been added.

In addition, I noted the absence of a more explicit connection to the concepts of secondary traumatization, trauma transmission, and compassion fatigue—topics highly pertinent to nursing students with clinical exposure during a pandemic. Given the emotional toll of such experiences, these constructs deserve at least some attention in the discussion and should ideally be situated in the theoretical background. Their inclusion would offer a more comprehensive understanding of the psychological landscape encountered by student nurses in times of crisis.

As you suggested, we have included these concepts in the “introduction.”

Finally, I would like to point out that new theoretical perspectives are occasionally introduced later in the manuscript (e.g., in the results or discussion sections) rather than being clearly established in the theoretical framework. To improve clarity and coherence, it would be advisable to relocate these conceptual discussions to the background section, thereby ensuring a more logically structured narrative throughout the article.

We have expanded the “introduction” to better link the theoretical framework with the concepts presented in ‘results’ and “discussion”.

In conclusion, while the study provides a relevant and important contribution to the field, its full value will only be realised through careful revision. I commend the authors for their efforts and encourage them to further develop the manuscript with these suggestions in mind.

More detailed feedback and comments:

I would like additional information on how the Spanish version of instruments has been provided. For example, did the authors provide the Spanish version of the Fear of COVID-19 scale themselves? How? Has this scale been validated and used in previous research yet? Is it consistent? (Cronbach Alpha score)

The instruments used have been adapted and validated for the Spanish population. The Spanish version of the Fear COVID-19 scale was obtained from the publication (Martínez-Lorca et al., 2020). This article describes the psychometric properties and wording of the questionnaire items. We have expanded the description of the instruments. We have also included the reliability coefficients (internal consistency) of the instruments in the sample used.

 It is not clear why the research decided to use and measure only 4 of the 5 NEO-FFI dimensions?

Data collection was very complicated in all the investigations carried out during the pandemic, especially in longitudinal studies. Our goal was to include as few items as possible in order to collect the information necessary for the research. The openness dimension of the NEO-FFI was eliminated because it has not typically been found to be a significant predictor of burnout. A comment is included in the manuscript.

For the different instruments, I would also have liked some examples of items in the text.

The measurement instruments are all validated questionnaires, the items of which we prefer not to reproduce. Some are well-known tests, and the lesser-known ones contain the wording of the items in the corresponding citation. We have expanded some of the descriptions of the questionnaires. Information on the psychometric aspects of the instruments in the sample used has also been included.

 Generally, I would appreciate some information about the validity and the reliability of the scales/instruments used in this study?

Information on psychometric aspects (indicators of validity and reliability) of the instruments used has been included.

For the elaboration of regression models, I assume that the authors made the choice of a stepwise approach in order to obtain the best model. How did they make the choice for different variables. Are the different predictor variables introduced in the model according to the theoretical hypotheses of the researchers?

The predictor variables in the regression models were selected based on their theoretical relevance according to the scientific literature on burnout syndrome in nursing staff. This is discussed in the data analysis section of the article.

Explain Table 1 in words. A table should not be seen as a replacement of text or explanation. It helps to synthetize but is it is not the job of the reader to extricate the necessary information from the table(s). The numerous abbreviations make this task even more complicated.

The information in the article has been expanded.

Please consider a different way to present the results more clearly.

The information in the work results section has been expanded and improved.

Table 2 mentions the model of Golembiewski? This is new for the reader since this model has not been presented previously in the theoretical section. As such, the is no information in a statement about stage 6 or 7 of the model. Also in the discussion section, there is nothing additional on this model.

The clarification regarding the model has been included in the article.

Table 3 also needs more explanation in order to make the results understandable for the reader.

To facilitate interpretation of the results, an explanation has been added to Table 3.

Discussion

What is the clinical relevance or usefulness of sentences like '... 18% were classed as probable causes of anxiety and 10,8% as liable to depression'?

The sentence that appears in the paper is not exactly the one indicated by the reviewer. The sentence in the discussion literally says the following: “18% were classed as probable cases of anxiety and 10.8% as liable to depression”. This sentence reports what has been obtained by applying the HADS. We have included in the description of the HADS the range of scores that reflect possible and probable cases of both anxiety and depression.

The authors should make the difference between 'symptoms that may be indicative of ... (diagnosis in DSM-5 or ICD-11), symptomatology that may be an indication of... but these statements contain no relevant clinical information. Therefore, already in the methods section, it is important to mention WHAT EXACTLY is measured. It should also be presented that clearly in the results section. Therefore, both sections still need a lot of improvement.

Thank you for your comment. We have included a brief clarification in section 2.2. We have also expanded the information in section 2.4. to provide a better understanding of what is being measured.

Reviewer 3 Report

Comments and Suggestions for Authors

Thank you for the opportunity to review this manuscript titled "Burnout and Professional Engagement during the COVID-19 Pandemic among Nursing Students with Clinical Experience: A Pre-Post Study." The manuscript presents an important and timely topic within nursing education and mental health during the COVID-19 crisis. However, substantial revisions are necessary before it can be considered for publication. My comments are as follows: 

Abstract part Revision Required: The abstract lacks a clear indication of the data collection period. It is crucial to specify that data were collected in February and June 2021, corresponding to the pre- and post-clinical internship phases. Suggestion: Clarify the timeline in the "Methods" sentence to improve transparency and temporal relevance. 

Introduction

Strengths: Provides a reasonable rationale for focusing on burnout and engagement in nursing students.

Areas for Improvement: The introduction would benefit from national-level data or literature specific to Spain on burnout prevalence among nurses or nursing students. Adding statistics or references from Spanish public health agencies or nursing associations could emphasize the local relevance of the topic. 

Methods – Instruments

Expansion Needed: While the study uses well-established instruments (e.g., HADS, NEO-FFI, UWES, FCV-19S), the description of each instrument is too brief. Suggestions:Report reliability coefficients (Cronbach’s alpha) from the current sample for each tool. Include information on the Spanish validation studies for these instruments, particularly for FCV-19S and the Granada Burnout Questionnaire. Clarify any cutoff values used for classifying levels of anxiety, depression, or burnout.

Ethical Considerations

Currently Insufficient. Ethical approval is mentioned too briefly.

Required Revision: Indicate the name of the ethical board or committee that granted approval (e.g., University of Granada Research Ethics Committee). Provide the approval number or protocol code, if available. Include the exact date of ethics approval. 

Discussion

General Strength: The discussion aligns well with the study aims and findings.

Enhancements Suggested: Enrich the discussion by including national-level data from Spain (e.g., from Ministry of Health reports, Consejo General de Enfermería) on mental health or burnout rates among student or professional nurses during COVID-19. Include comparative international evidence, such as similar studies conducted in Europe, Latin America, or Asia, to contextualize the findings. Offer practical recommendations for nursing education policy or clinical internship planning during health crises.

Study Limitations

Please expand the limitations section: Use of non-probability sampling, Potential recall or social desirability bias, as the same students completed both waves, No qualitative data to explore contextual influences on stress and coping, Single-institution sample limits generalizability.

Online Data Collection: The use of online self-reported questionnaires, while necessary due to pandemic restrictions, may limit the reliability of responses. Factors such as lack of supervision, environmental distractions, and variable digital literacy can affect the accuracy and completeness of the data. Additionally, online methods may restrict participation from students who are less motivated or experiencing high levels of distress, potentially leading to selection bias.

Acknowledgements

Add a brief note of appreciation to the participating students, given the extraordinary demands and uncertainty of the COVID-19 period. 

Carefully proofread for consistent use of terms (e.g., “concientiousness” should be corrected to “conscientiousness”). Ensure that all abbreviations are defined on first use. Consider including a visual flowchart or timeline of the study design and data collection for clarity.

Author Response

Thank you for the opportunity to review this manuscript titled "Burnout and Professional Engagement during the COVID-19 Pandemic among Nursing Students with Clinical Experience: A Pre-Post Study." The manuscript presents an important and timely topic within nursing education and mental health during the COVID-19 crisis. However, substantial revisions are necessary before it can be considered for publication. My comments are as follows:

Abstract part Revision Required: The abstract lacks a clear indication of the data collection period. It is crucial to specify that data were collected in February and June 2021, corresponding to the pre- and post-clinical internship phases. Suggestion: Clarify the timeline in the "Methods" sentence to improve transparency and temporal relevance.

We have already completed the abstract part.

Introduction

Strengths: Provides a reasonable rationale for focusing on burnout and engagement in nursing students.

Areas for Improvement: The introduction would benefit from national-level data or literature specific to Spain on burnout prevalence among nurses or nursing students. Adding statistics or references from Spanish public health agencies or nursing associations could emphasize the local relevance of the topic.

We have expanded the justification by adding studies in Spain and data from official sources.

Methods – Instruments

Expansion Needed: While the study uses well-established instruments (e.g., HADS, NEO-FFI, UWES, FCV-19S), the description of each instrument is too brief. Suggestions:Report reliability coefficients (Cronbach’s alpha) from the current sample for each tool. Include information on the Spanish validation studies for these instruments, particularly for FCV-19S and the Granada Burnout Questionnaire. Clarify any cutoff values used for classifying levels of anxiety, depression, or burnout.

More information has been included in all the measurement instruments used. The FCV-19S now includes the range of scores used to classify subjects. All information about this instrument, including the wording of the items and psychometric aspects, can be found in (Martinez et al., 2020). Information about the Granada Burnout Questionnaire has also been expanded. This instrument is the property of the authors of this work, so we are making all the information we have about the test available to reviewers or any interested researchers.

Ethical Considerations

Currently Insufficient. Ethical approval is mentioned too briefly.

Required Revision: Indicate the name of the ethical board or committee that granted approval (e.g., University of Granada Research Ethics Committee). Provide the approval number or protocol code, if available. Include the exact date of ethics approval.

The registration number and date of approval by the ethics committee have been added.

Discussion

General Strength: The discussion aligns well with the study aims and findings.

Enhancements Suggested: Enrich the discussion by including national-level data from Spain (e.g., from Ministry of Health reports, Consejo General de Enfermería) on mental health or burnout rates among student or professional nurses during COVID-19. Include comparative international evidence, such as similar studies conducted in Europe, Latin America, or Asia, to contextualize the findings. Offer practical recommendations for nursing education policy or clinical internship planning during health crises.

We have expanded the “discussion” to include prevalence data for nursing students and registered nurses following the pandemic. Information from two meta-analyses covering a wide range of countries is included. In addition, we have expanded on this information by discussing the importance of prevention strategies, with examples of how students could develop personally and professionally within their academic training.

Study Limitations

Please expand the limitations section: Use of non-probability sampling, Potential recall or social desirability bias, as the same students completed both waves, No qualitative data to explore contextual influences on stress and coping, Single-institution sample limits generalizability.

We have extended the limitations according to your instructions.

Online Data Collection: The use of online self-reported questionnaires, while necessary due to pandemic restrictions, may limit the reliability of responses. Factors such as lack of supervision, environmental distractions, and variable digital literacy can affect the accuracy and completeness of the data. Additionally, online methods may restrict participation from students who are less motivated or experiencing high levels of distress, potentially leading to selection bias.

The students completed the questionnaire on the web platform but did so in person, at the end of face-to-face seminars and under supervision. We have included this information in the manuscript (2.3. Procedures).

Acknowledgements

Add a brief note of appreciation to the participating students, given the extraordinary demands and uncertainty of the COVID-19 period.

We have added a thank you note as requested.

Carefully proofread for consistent use of terms (e.g., “concientiousness” should be corrected to “conscientiousness”). Ensure that all abbreviations are defined on first use. Consider including a visual flowchart or timeline of the study design and data collection for clarity.

Sorry for the mistakes. It's fixed now. We have added a visual flowchart.

Reviewer 4 Report

Comments and Suggestions for Authors Congratulations to the authors for their work. However, some major corrections are needed: There are several articles in the literature on burnout and engagement during the pandemic. I understand that this is a post-study that follows on from another previously published study, but the difference in this study should guide the publication.
I have some reservations:
ABSTRACT Include the study design, data collection period and analysis method. Include that it was derived from another publication. In the conclusion, reformulate it to include the impacts of the research on the scientific community. INTRODUCTION
The study's differences with other published studies should be included, both those by the same author and others. Make it clear that if you intend to compare clinical experience with burnout and engagement, there are other studies carried out with trained professionals. METHOD It is necessary to include the study design. 2.1 Include the criteria for participants - remember that this is another article and sometimes the reader has not read the first one already published. 2.2 Include who these research members are.
In the study already published there are 212 participating students and in this one 112. Why was there such a significant loss? Include It is not clear what was provided as an intervention. Was it only clinical experience? Was the use of seminars mentioned... how was it done? Was it on the topic? Time? Was it a seminar followed by an internship?
2.3 Include how the data was collected, in person or remotely, if it was at the university and how many questions there were.
RESULTS Ok But, I think that item 3.4, which addresses the study objectives, is very shallow and not very clear. It should include more data. Are there more findings about it? DISCUSSION Include the main pedagogical strategies from the literature to minimize the impact of burnout. There is evidence that speaks about strategies that universities can include in their training curricula. CONCLUSION Deepen and discuss the impacts of research on nursing, the health sector and the scientific community. Appropriate citations

Author Response

Congratulations to the authors for their work. However, some major corrections are needed: There are several articles in the literature on burnout and engagement during the pandemic. I understand that this is a post-study that follows on from another previously published study, but the difference in this study should guide the publication.

I have some reservations:

ABSTRACT Include the study design, data collection period and analysis method. Include that it was derived from another publication. In the conclusion, reformulate it to include the impacts of the research on the scientific community.

We have already added this information.

INTRODUCTION

The study's differences with other published studies should be included, both those by the same author and others. Make it clear that if you intend to compare clinical experience with burnout and engagement, there are other studies carried out with trained professionals.

We have included additional bibliography to justify the study of burnout and work engagement in the “introduction”.

METHOD It is necessary to include the study design.

The design has been included in section 2.2.

2.1 Include the criteria for participants - remember that this is another article and sometimes the reader has not read the first one already published.

The information in the “participants” section has been expanded.

2.2 Include who these research members are.

In the study already published there are 212 participating students and in this one 112. Why was there such a significant loss? Include It is not clear what was provided as an intervention. Was it only clinical experience? Was the use of seminars mentioned... how was it done? Was it on the topic? Time? Was it a seminar followed by an internship?

Data collection was very complicated in all the research carried out during the pandemic. In longitudinal studies, a loss of subjects is to be expected, but it is true that in this case, the pandemic situation made it more complicated than usual. Data collection was scheduled during the students' face-to-face classes, but it is possible that many of them were at home or in other circumstances that contributed to the loss being much greater than expected.

The “intervention” would be the clinical experience acquired during the internship period that the students were beginning. It was an informative seminar about the internships they were about to begin, in which the questionnaires were administered through the students' access to the online platform prepared for this purpose.

An explanatory comment has been included in the paper.

2.3 Include how the data was collected, in person or remotely, if it was at the university and how many questions there were.

The information was collected in person during seminars that complemented the healthcare placements. This was the only time that students attended the university in person. An online platform was used for this purpose, with each student individually completing a total of 142 items.

RESULTS Ok But, I think that item 3.4, which addresses the study objectives, is very shallow and not very clear. It should include more data. Are there more findings about it?

The work includes all the information obtained about the variables included. The information relating to all the variables recorded has been expanded and the objectives of the study have been written up in a more complete manner. The entire results section has been expanded, as suggested.

DISCUSSION Include the main pedagogical strategies from the literature to minimize the impact of burnout. There is evidence that speaks about strategies that universities can include in their training curricula.

We have expanded on this information by discussing the importance of prevention strategies, with examples of how students could develop personally and professionally within their academic training.

CONCLUSION Deepen and discuss the impacts of research on nursing, the health sector and the scientific community. Appropriate citations

References should not be included in the “conclusions.” However, we have included new information at the end of the “discussion” with appropriate citations. This information reflects the need for prevention and intervention strategies and the potential impact on the nursing community.

Reviewer 5 Report

Comments and Suggestions for Authors

The manuscript follows a recognizable structure and is written in a good academic style. On the other hand, there are a lot of elements that require more attention and improvement.

1. The title and research results are not aligned. The title clearly indicates two variables: Burnout and Professional Engagement. On the other hand, in section 2.3. Instruments lists a large number of variables, as in Table 1. It is necessary to clearly define the research focus.

2. What is the goal and purpose of the manuscript? What is a research gap? What is new about the manuscript? This should be clearly described and defined at the end of the Introduction section.

3. The Results section lacks a check of the validity of the measurement scales. Data for testing reliability and Cronbach's alpha for each measurement scale are missing.

4. Missing information on how the sample size was determined? What is the evidence that the sample is representative? The description and argumentation should be described in detail and presented in the Participant section.

5. In the Results section it is written as follows "Hypothesis tests were conducted to identify possible differences between groups of participants, for each of the variables considered.", as well as "Hypothesis tests were also performed to detect differences between the variables according to the impact produced by COVID-related factors." What are the research hypotheses?

6. In the Discussion section the limitations are listed, however there are many more limitations than those listed, which are very serious. A fair approach and a clear presentation of the objective limitations of the study are expected.

7. The list of references seems very modest. It is necessary to include more sources in the list of references, especially because there are many previous articles with related content.

The manuscript generally presents the topic and results of the study which is quite outdated.

Author Response

The manuscript follows a recognizable structure and is written in a good academic style. On the other hand, there are a lot of elements that require more attention and improvement.

Dear reviewer, thank you for your comments. Below we have described the changes made.

  1. The title and research results are not aligned. The title clearly indicates two variables: Burnout and Professional Engagement. On the other hand, in section 2.3. Instruments lists a large number of variables, as in Table 1. It is necessary to clearly define the research focus.

The title has been completed. The requested clarification has been included at the end of the “introduction,” in “methods,” and “results” of the research.

  1. What is the goal and purpose of the manuscript? What is a research gap? What is new about the manuscript? This should be clearly described and defined at the end of the Introduction section.

We have rewritten this part of the introduction.

  1. The Results section lacks a check of the validity of the measurement scales. Data for testing reliability and Cronbach's alpha for each measurement scale are missing.

This information has been included in the measuring instruments.

  1. Missing information on how the sample size was determined? What is the evidence that the sample is representative? The description and argumentation should be described in detail and presented in the Participant section.

No sample was selected from among the students who met the requirements to participate in the research. All third-year nursing students at the University of Granada who met a single condition participated: they were completing their clinical practicum. Initially, 212 students participated. This research included the 112 students who correctly completed the questionnaires at both measurement points.

  1. In the Results section it is written as follows "Hypothesis tests were conducted to identify possible differences between groups of participants, for each of the variables considered.", as well as "Hypothesis tests were also performed to detect differences between the variables according to the impact produced by COVID-related factors." What are the research hypotheses?

All hypothesis tests cited in the “Data Analysis” section are statistical hypothesis tests. No prior research hypotheses were proposed due to the uncertainty generated by the pandemic situation in which we found ourselves. The term “Statistical” has been added to the paper.

  1. In the Discussion section the limitations are listed, however there are many more limitations than those listed, which are very serious. A fair approach and a clear presentation of the objective limitations of the study are expected.

We have rewritten the limitations part of the discussion.

 The list of references seems very modest. It is necessary to include more sources in the list of references, especially because there are many previous articles with related content.

We have expanded the number of updated references to complement our results.

The manuscript generally presents the topic and results of the study which is quite outdated.

Thank you for your comment.

Round 2

Reviewer 3 Report

Comments and Suggestions for Authors

Dear Authors,

I would like to congratulate you on your work. This manuscript addresses a highly relevant and sensitive issue for nursing students during the COVID-19 pandemic. The pre-post design, use of validated instruments, and comparison with earlier data from the same student cohort strengthen the quality and originality of your study. Your findings, particularly the identification of risk profiles and the protective role of engagement, provide valuable insights that will contribute to both the academic literature and practical interventions in nursing education. I believe your work will inspire further research and inform strategies to better support students’ mental health and professional development. Thank you for your contribution to the field.

Author Response

Thanks for your comments. We appreciate it. 

Reviewer 4 Report

Comments and Suggestions for Authors

Congratulations on the revisions; the article has notably improved. However, I still have a few remarks:

  • In the introduction, the unique contribution or rationale for conducting the study is not yet clearly articulated.

  • In the methods section, the study design should be presented first (prior to the description of participants), as this is where you begin to specify the type of study. Additionally, please clarify whether it is a quantitative longitudinal design.

  • All other items have been fully addressed.

Author Response

Congratulations on the revisions; the article has notably improved. Thanks for your comments. We appreciate it.

However, I still have a few remarks:

  • In the introduction, the unique contribution or rationale for conducting the study is not yet clearly articulated. We have rewritten it. 

  • In the methods section, the study design should be presented first (prior to the description of participants), as this is where you begin to specify the type of study. Additionally, please clarify whether it is a quantitative longitudinal design. We have made these changes.

  • All other items have been fully addressed. Thanks!

Reviewer 5 Report

Comments and Suggestions for Authors

No additional comments.

Author Response

We appreciate it.